# Machine Learning to Predict the Progression of Bone Mass Loss Associated with Personal Characteristics and a Metabolic Syndrome Scoring Index

**DOI:** 10.3390/healthcare9080948

**Published:** 2021-07-28

**Authors:** Chao-Hsin Cheng, Ching-Yuan Lin, Tsung-Hsun Cho, Chih-Ming Lin

**Affiliations:** 1Division of Chest Medicine, Ten-Chan General Hospital, Chung Li, Taoyuan 320, Taiwan; starcheng2001@gmail.com; 2Department of Laboratory Medicine, Ten-Chan General Hospital, Chung Li, Taoyuan 320, Taiwan; lab02@tcmg.com.tw; 3Institute of Biomedical Informatics, National Yang-Ming-Chiao-Tung University, Taipei 112, Taiwan; eric101784@hotmail.com; 4Department of Healthcare Information and Management, Ming Chuan University, Taoyuan 333, Taiwan

**Keywords:** osteopenia, metabolic syndrome, socioeconomic status, lifestyle, machine learning

## Abstract

A relationship exists between metabolic syndrome (MetS) and human bone health; however, whether the combination of demographic, lifestyle, and socioeconomic factors that are associated with MetS development also simultaneously affects bone density remains unclear. Using a machine learning approach, the current study aimed to estimate the usefulness of predicting bone mass loss using these potentially related factors. The present study included a sample of 23,497 adults who routinely visited a health screening center at a large health center at least once during each of three 3-year stages (i.e., 2006–2008, 2009–2011, and 2012–2014). The demographic, socioeconomic, lifestyle characteristics, body mass index (BMI), and MetS scoring index recorded during the first 3-year stage were used to predict the subsequent occurrence of osteopenia using a non-concurrence design. A concurrent prediction was also performed using the features recorded from the same 3-year stage as the predicted outcome. Machine learning algorithms, including logistic regression (LR), support vector machine (SVM), random forest (RF), and extreme gradient boosting (XGBoost), were applied to build predictive models using a unique feature set. The area under the receiver operating characteristic curve (AUC), accuracy, sensitivity, specificity, precision, and F1 score were used to evaluate the predictive performances of the models. The XGBoost model presented the best predictive performance among the non-concurrence models. This study suggests that the ensemble learning model with a MetS severity score can be used to predict the progression of osteopenia. The inclusion of an individual’s features into a predictive model over time is suggested for future studies.

## 1. Introduction

Osteoporosis is a systemic bone disease and an important public health problem because it increases the incidence and mortality of fractures and significantly increases the risk of fracture-related medical expenses [1,2,3]. A recent review study reported that the economic burden of osteoporosis-related fractures was significant, costing approximately USD 17.9 billion and GBP 4 billion per annum in the USA and UK, respectively [4]. In Taiwan, the prevalence of osteoporosis among the population older than 50 years increased from 17.4% in 2001 to 25.0% in 2011 [5] Approximately one-quarter of individuals older than 65 years who have been diagnosed with osteoporosis have experienced a spine or hip fracture [6].

Several physical factors have been associated with osteoporosis, including abdominal obesity, high blood pressure, dyslipidemia, and glucose metabolism abnormalities, which are all considered to be components of metabolic syndrome (MetS). Cardiovascular diseases (CVDs) have been linked to reduced bone mineral density (BMD), osteoporosis, and osteopenia [7,8,9,10]. While MetS may play a potential role in the development of osteoporosis, further research is needed to obtain hard data to support the hypothesis. Previous studies have identified similar risk factors and pathophysiological mechanisms underlying the development of both osteoporosis and atherosclerotic CVDs. There are suggestions that common underlying pathways, such as disturbed calcium homeostasis, induction of inflammatory response, and oxidative stress, are shared by the two conditions. It has been suggested that the two conditions share underlying pathways linking components of MetS as well as the coupling process of bone formation and bone reabsorption [11,12]. Evidence suggests that consideration should be given to the correction of MetS for the prevention of osteoporotic fractures [13]. Potential factors that affect MetS development have included demographic factors (including age, sex, and living area) and lifestyle behaviors (including smoking, alcohol consumption, diet, and physical activity) [14,15]. Socioeconomic status (SES) components, including income, occupation, and education, are also closely related to CVD development and metabolic indicators [15,16,17]. Previous analyses may have been limited by the lack of inclusion of social and lifestyle covariate factors, which may reduce the explanatory power of these analyses. To clarify the causal relationship between bone density and MetS, a prospective longitudinal study should be performed, and during the investigation, sex, age, ethnicity, lifestyle, and eating habits should not be overlooked [13,18].

For decades, artificial intelligence has been applied to the identification of risk factors or groups at risk of developing osteoporosis. The burden on health systems, the economy, and society could be lessened through the use of an artificial intelligence model to predict risk groups [19,20,21,22]. A comprehensive and low-cost method could be developed to facilitate the use of predictive models during health examinations, especially for developing countries or rural areas. However, most predictions for osteoporosis have been modeled using information for participants who have primarily been female or in specific age groups. Predictive tools should be developed to perform similarly across various populations, including greater numbers of participants across a large age range, which has not been the case for existing predictive models [22]. Additionally, few studies have performed predictive models to identify the risk for osteopenia, which represents an earlier stage of bone disorders, and to identify those at risk of osteopenia that may be useful for promoting overall bone health, especially among younger populations.

To better understand the relationship between MetS and human bone health, determining whether the underlying demographic, lifestyle, and socioeconomic causes of MetS also affect bone density is critical. Our study aimed to explore a comprehensive approach applicable to a wider population. Recent studies have reported that MetS severity scores can serve as useful indicators to assess the potential risk factors for subclinical conditions and can facilitate the development of prevention strategies during the early stages of disease development [23,24]. To our knowledge, no study has previously developed a disease prediction model that combines demographic, lifestyle, and SES with MetS score indicators. Except for the two types of research that we reported [15,21], most studies [14,16,17,18,19,20] conducted a cross-section (i.e., concurrence) approach to modeling the potential risks associated with bone health. A non-concurrence study examining these factors can be used to investigate the causal relationship between these factors and bone health. Using a machine learning approach, this study developed a model to predict the loss of adult bone mass among a Taiwanese population using MetS severity scores and individual risk factors.

## 2. Materials and Methods

### 2.1. Data Source

Data were obtained and analyzed from a membership-oriented private institute, a Major Health Screening Center in Taiwan. With four clinic locations around the country, the center provides periodic health examinations to approximately 600 thousand members. A detailed description of the data collection and analysis of the resulting Major Longitudinal Health-check-up-based Population Database (MJLPD) is described in detail elsewhere [25,26].

In consideration of various ethical issues within the data, the protocol of this study was evaluated and approved by the National Taiwan University Research Ethics Committee (NTU-REC 201911EM012) and the Major Health Screening Center.

### 2.2. Study Sample

Participants over 20 years of age who had undergone at least one standard health screening at the Center in each of three three-year stages/periods (i.e., 2006–2008, 2009–2011, and 2012–2014) from 2006 to 2014 were used to conduct the longitudinal study. All participants lacking BMD examination data or who were diagnosed with osteopenia or osteoporosis (T score < −1) at baseline (i.e., 2006–2008) were excluded from the study. For participants who had undergone multiple screenings within the three-year period, the last examination period was selected for the analysis. As a result, three questionnaires and examination measurements for each participant were collected during the nine-year period. A final total of 23,497 participants (13,012 males and 10,485 females) met the inclusion criteria and were used as our study dataset. Among the included study population, 1402 and 1805 participants were diagnosed with either osteopenia or osteoporosis during the second and third stages, respectively. Due to a relatively low positive rate, the dataset was analyzed using a random under-sampling (1:1 match) approach while applying machine learning models to mitigate the imbalance problem. A flow chart of the data collection process used to identify the study participants and define the analysis dataset is shown in Figure 1.

### 2.3. Response Variables

The measurement of BMD in this study was primarily performed using a Lunar DPX-L density meter, which measures dual-energy X-ray absorption (Liberty Corp., Madison, WI, USA). Using the National Health and Nutrition Examination Survey as a reference population, gender-specific T scores were calculated, and osteoporosis was defined as a T score below −2.5 standard deviations (SDs) relative to the average population value, whereas a T score between −1.0 and −2.5 SDs was defined as low BMD (referred to as osteopenia), and a T score above −1.0 SD was defined as normal [27]. All BMD reports were independently reviewed and coded by trained research physicians. Bone measurements taken at the spine were given priority, followed by hip bones and wrist bones, and the results of all measurement sites were considered by physicians. In conducting the study of the effects of risk factors on bone health over an extended period, we collected the indicators of ongoing osteopenia or osteoporosis status among those who developed these disorders during the study period and were not diagnosed with bone disorders at baseline. Using −1.0 SD as the cutoff point in the current longitudinal study, individual BMD was treated as a dichotomous variable. The measured outcome was defined as the occurrence of bone illness, as diagnosed during the second and/or third stages for those with BMD values higher than −1.0 SD during the baseline measurement.

### 2.4. Explanatory Variables

Each of the study participants completed a self-administered questionnaire during screening to obtain socio-demographic characteristics and lifestyle habit information. Data collected included sex, age (classified into 20–39 years, 40–64 years, and 65 or more years), four aspects of SES (i.e., marriage, education, income, and occupation), as well as nine well-documented lifestyle habits, constituting related risk factors in past studies. Hormones, steroids, and thyroid-related treatment drugs used by patients were cataloged. 

Body mass index (BMI, kg/m^2^) is considered a risk factor for osteopenia and was included as a continuous variable in our analysis. Using the same databases reported in previous research [24], by back-transforming the standardized scores derived from the aforementioned equations, a covariance matrix was obtained with the MetS severity scores, calculated using waist circumference, fasting plasma glucose, systolic blood pressure, fasting triglycerides (TG), and high-density lipoprotein (HDL). First, the individual values of the five components were standardized and converted to a Z score. A confirmatory factor analysis approach was then followed to derive the score based on the five MS components, with a weighted contribution for each of the components to a latent MetS factor being determined based on both specific age ranges and genders. In the present study, a higher score denotes that a person has a more severe MetS condition, whereas lower scores indicate the lack of MetS.

Classification of the SES of participants was divided into three levels of educational attainment. Occupation was classified as unemployed, manager/owner, and non-management employee. Marriage classification was labeled as married or unmarried. Some lifestyle habits, estimated quantitatively by frequency, were classified into three levels for smoking, alcohol consumption, sugar-sweetened consumption, physical activity, and sleeping. Other habits, such as betel nut chewing, vegetarian diet, dairy intake, taking calcium supplements, and related medical treatments such as hormones, steroids, and thyroid-associated medications, were classified dichotomously. 

### 2.5. Study Design

The physical and biochemical aspects, demographic, socioeconomic, and lifestyle characteristics of the study participants at baseline and those associated with participants who developed osteopenia or osteoporosis over the following two stages were described. In the longitudinal study, the causal relationships were analyzed using a non-concurrent design. The lifestyle characteristics, demographic, socioeconomic, and, as well as BMI and MetS scores from the first stage were used as the features recorded during the following two stages. The factors used to predict the occurrence of osteopenia or osteoporosis for the second and third stages were those identified during the initial stage. A concurrent prediction was also performed during the second and third stages using each individual’s features from the stage being examined.

### 2.6. Feature Selection and Machine Learning

All features missing greater than 30% of values were excluded from the analysis. The missing values for the remaining features were replaced by using a multiple imputation technique. A multivariate imputation via chained equations (MICE) module was used in the R package to perform the data imputation. To identify the effects of important features on the development of osteopenia, we applied the random forest (RF) algorithm with 10 times repeated 10-fold cross-validation to select robust significant features from the training/validation (80/20) dataset, which utilized a mixture of numerical and categorical features. Both the features and the cutoff points associated with each feature were randomly chosen before each training model. Thus, the sequence of feature importance could differ during each model. Then, we averaged the ten lists of feature importance to obtain a robust selected feature list. The results demonstrated that the independent variables for forecasting the prediction included 17 of the 24 analyzed features, which were selected as a selected features dataset for further machine learning and model evaluation. The MetS score and BMI played the most important roles among the selected features (Table 1).

In this study, four well-accepted machine learning algorithms, including logistic regression (LR), extreme gradient boosting (XGBoost), RF, and support vector machine (SVM), were applied to develop the concurrence and non-concurrence predictive models. A 10-fold cross-validation and grid search were used to determine the parameters of the four predictive models for the tuning of hyperparameters while training the model using the defined dataset. Using bootstrapping and 10-fold validation, the best scores were used to define the parameters for the predictive models. The test of a dataset with 80/20, which was a separated dataset from the preceding datasets, was used to avoid the development of an over-fitting model. Subsequently, 10 iterations of a receiver operating characteristic curve (ROC) analysis were employed on the randomized datasets to obtain the best area under the ROC curve (AUC). All machine learning analyses were performed using Python Software (Foundation and Python Language Reference, version 3.7.3, Beaverton, OR, USA). The libraries of Scikit-Learn 0.23.2 were implemented and used to confirm these models. The process used for feature selection and machine learning is shown in Figure 2.

### 2.7. Model Evaluation

The model’s discrimination was measured. In this study, discrimination refers to the predictive effectiveness of the model in determining between participants with and without osteopenia. In each model, the discriminatory power was analyzed based on the AUC, while the ROC curves used were determined by plotting the true positive fraction against the false positives. For each cutoff score, the specificity (maximum subsequent sum) and sensitivity (optimal values) were calculated.

Furthermore, accuracy, precision, and F1 score evaluation indicators from the confusion matrix were used to analyze the relationship between the actual values and the predicted values for osteopenia. The precision–recall curve (PRC) was also generated to determine the tradeoff between precision and recall at different thresholds. Precision–recall is a useful measure of the success of prediction when classes are imbalanced. In the imbalanced data, the false-positive rate tends to stay at small values due to the low positive rate. Thus, ROC becomes less informative for the model performance in this situation. On the other hand, the PRC baseline is varied by the value of the positive rate, and PRC is performed by switching from false positives to precision, which provides more valuable information. A high AUC represents both high recall (i.e., sensitivity) and high precision, where high precision is associated with a low false-positive rate, and high recall relates to a low false-negative rate. F1 was calculated as 2 × precision × recall/(precision + recall), which is the harmonic mean of precision and recall. A larger F1 score indicates a more accurate model.

## 3. Results

Lifestyle habits, sociodemographic factors, and biochemical and physical examination items over the three study stages are presented in Table 2. Osteopenia occurrence rates during the second stage in men and women were 7.3% and 4.3%, respectively. An increase to 8.3% and 6.9% in the occurrence of osteopenia was observed during the third stage. Participants with relatively low SES, adverse habits (e.g., smoking and alcohol consumption), low sleep hours, and a vegetarian diet and who were taking related medicines during the initial baseline stage had a higher occurrence of osteopenia during the subsequent stages. Compared with the participants at baseline, the average BMI of those who developed osteopenia in subsequent stages was relatively low. The participants who went on to develop osteopenia had a lower MetS score (−0.22) in the subsequent stages compared with the average score (0.09) for the entire population at baseline.

The study utilized four machine learning models (i.e., LR, XGBoost, RF, and SVM) to predict osteopenia. The predictive models were corroborated using optimal parameters for each model through a grid search. The ROC and PRC curves of the generated machine learning models for the concurrence and non-concurrence designs are shown in Figure 3 and Figure 4. 

The differences between the models were more distinct when using baseline features to predict osteopenia during the second stage than when using the baseline features to predict osteopenia during the third stage. The performances of the models for predicting osteopenia occurrence during the second stage according to baseline features are shown in Table 3. 

The AUC values for LR, XGBoost, RF, and SVM in the non-concurrence and concurrence models were 0.726 and 0.745, 0.753 and 0.721, 0.693 and 0.687, and 0.723 and 0.712, respectively. The F1 scores for the four algorithms in the non-concurrence and concurrence models were 0.668 and 0.689, 0.723 and 0.686, 0.656 and 0.633, and 0.688 and 0.676, respectively. Among all predictive models, the XGBoost model had the highest AUC value. Except for the LR models, most of the non-concurrence models demonstrated better predictive performance than the concomitant concurrence models. Although the concurrence LR model was associated with high AUC and PRC values of 0.745 and 0.774, respectively, the other indicators were relatively poor. The performance of the predictive models for identifying osteopenia occurrence during the third stage showed a similar pattern, with poorer performances than the models used to predict occurrence during the second stage (Table 4).

## 4. Discussion

Most previous studies have been conducted using a concurrence design, also known as cross-sectional design, which does not allow for the assessment of causal relationships between risk factors and BMD. By initially selecting participants without osteopenia and using a prospective dataset, the present study indicates that non-concurrent models resulted in better predictive performance and are more suitable for this empirical purpose than concurrent models while the optimal algorithm (i.e., XGBoost) is being applied. Therefore, further investigation remains necessary to verify these findings, especially for chronic disorders such as osteopenia or osteoporosis. In addition to BMI, the MetS severity score is identified as the dominant predictor of osteopenia in the present study. Though a relationship has been explored between MetS and bone health, some confusion may arise from the traditional Adult Treatment Panel criteria, such as whether individuals with two high-level MetS components have a lower CVD risk than in those with slightly elevated levels above the criteria in three or more components. Due to the limitations in the traditional MetS criteria, we instead developed the models with a MetS severity score to provide valuable evidence for healthcare societies. Additionally, the study found better predictive performance for the second stage than for the third stage, which implies that the selected features are suitable for predicting osteopenia occurrence over the short-term period of three years but may not be suitable for predictions over a longer period. It could be justified that there would be less effects of health outcomes because of even early socioeconomic or behavioral conditions since these conditions may have changed overtime due to certain personal or heath issues. In the past, risk calculators, such as the web-based Fracture Risk Assessment Tool (FRAX^®^) algorithm, have enabled the assessment of an individual’s fracture risk using clinical risk factors, such as age and alcohol consumption [28]. A prediction of osteopenia using easily measured risk factors may alert practitioners to the condition of an individual’s bone health during the early stages of bone disease and may enhance the performance of osteoporosis prevention or avoid the occurrence of future fractures. Our findings may encourage health institutes to provide prevention strategies to those who are potential osteopenia patients, which will lead to better bone health in over one thousand people or the possibility of avoiding deterioration in advance for the study participants. The results for a field with limited research provide pertinent and comprehensive information to those who seek to identify the most suitable model in bone mass loss for decision-making.

Predictive algorithms can serve as diagnostic screening tests to stratify patient populations by risk and to allow for more discrete decision-making [29]. Since screening is intended to guide interventions, high accuracy and precision testing is required. We applied four machine learning algorithms to the construction of predictive models. Generally, RF and XGBoost are ensemble learning models, and LR represents the basic machine learning model, while SVM is widely used as a predictor. As previously discussed, ensemble learning models, specifically the XGBoost model, was found to have higher prediction capabilities and lower risks of overfitting than the others, which can provide greater benefits to decision-makers who are looking for more suitable models for the prediction of healthcare demands [30]. Cruz et al. conducted a review study and summarized the different performances of various machine learning-based diagnostic models for osteoporosis among 25 studies, taking into account the artificial intelligence method applied, the number of risk factors included in the model, the number of patients evaluated, the country associated with the evaluation, and the proportions of each sex in the study population [22]. The study noted that most of the proposed systems can be very useful for the medical community, provided that analysis is not restricted to specific groups and that a spectrum of input variables is included. To the best of our knowledge, this study is the first to develop and compare various machine learning models to predict early bone mass loss that also considers socioeconomic and lifestyle conditions, in addition to MetS indicators.

Compared with linear-based models, neural networks constitute flexible nonlinear systems and may be more suitable for the prediction of outcomes when the associations between the variables are nonlinear, complex, and multidimensional, as is done when assessing the relationships among variables in complex biological systems [31]. Using neural networks, de Cos Juez et al. studied the influence of diet and lifestyle on BMD values in postmenopausal women. A questionnaire examining nutritional habits and lifestyles was used, resulting in 39 variables, such as calcium intake, protein intake, number of pregnancies, height, and BMI, for each respondent [19]. They found that these variables influenced the progression of osteoporosis. However, collecting all possible individual predictors can be difficult, and not all predictors apply to routine disease prevention. To reduce the number of input variables required to obtain an accurate predictive model, the researchers further processed the identified variables using genetic algorithms, which resulted in a model that demonstrated better performance. To test the performance of the algorithm, we performed artificial neural network models utilizing the same dataset via the machine learning module in the SAS Viya Plus package (Linux^®^ for x64, SAS Institute Inc., Cary, NC, USA). The results showed similar performance (e.g., AUC = 0.732 for the non-concurrence model for the second stage) as that obtained for the present LR model. However, the current performance of the predictive models developed in the present study still has room for improvement. In particular, there are even lower (AUC < 0.65) performances when only the two most significant features (i.e., MetS score and BMI) are being used to predict. Other than the modelling strategy that the study used, there are several different algorithms (e.g., gradient boosting machine, decision tree, etc.), samplings, and feature selections such as data-driven approaches [32,33,34] that have been developed. As we performed additional analyses under varying approaches, the results showed that the synthetic minority oversampling technique could be used to optimize the performance of machine learning (Appendix A). Future studies with the approach may be applicable. However, caution should be exercised to prevent adding increased uncertainty, especially in regard to a sample with a small number of examples of a minority class or a non-continuous feature space [35]. Additionally, Loke et al. studied the association between MetS and BMD and found that the correlation had a very different effect among men than among women [18]. Despite considering the effects of sex and age and using sex- and age-based MetS severity scores in the present study, subgroup analyses stratified by sex and age might provide more information in future studies.

Various issues, including patient self-selection, confounding due to various indications, and the inconsistent availability of outcome data, can result in the inadvertent introduction of bias in machine learning-based predictions [29]. The present study has some limitations that must be addressed. First, although the use of several medications was considered, information regarding some treatments related to BMD was not available, including treatments associated with chronic renal insufficiency, bone metabolic illness, chronic hepatopathy, and neoplasia. A recent study suggested that genomic data can be used to develop a predictive model for BMD using a machine learning approach [36]. However, genotypic variables were not collected in our study, which may have impacted the performance of bone mass prediction models. Additionally, the issues of information loss and selection bias raised by the under-sampling method and imputation procedure cannot be ignored. Finally, the concept of social mobility refers to the degree of SES stability or change in the trajectory of an individual’s life course itself. Therefore, exposure to socioeconomic or behavioral adversities during the course of life increases an individual’s long-term risk. The accumulation of risk models advocates that increased exposure, duration, and severity to adverse events during the life course increase the risk of disease development [37]. The development of a life course approach has been suggested to develop a better understanding of how a reciprocal relationship between affected factors and health changes over different life stages [38].

## 5. Conclusions

The study found that an individual’s MetS severity score, BMI, and socioeconomic and lifestyle indicators could be used as tools to predict the progression of bone density health using an ensemble learning model. The prediction of osteopenia using easily measured risk factors may alert a physician to the precarious condition of an individual’s bone health during the early stages of bone disease and may enhance the performance of preventative measures to avoid osteoporosis or further fractures, reducing the economic burdens associated with related diseases. Our findings can provide guidance for health care providers when designing health promotion measures for specific populations. However, to reflect real-world conditions, the inclusion of an individual’s specific features into a predictive model, including changes that occur over time, is suggested for future studies.

## Figures and Tables

**Figure 1 healthcare-09-00948-f001:**
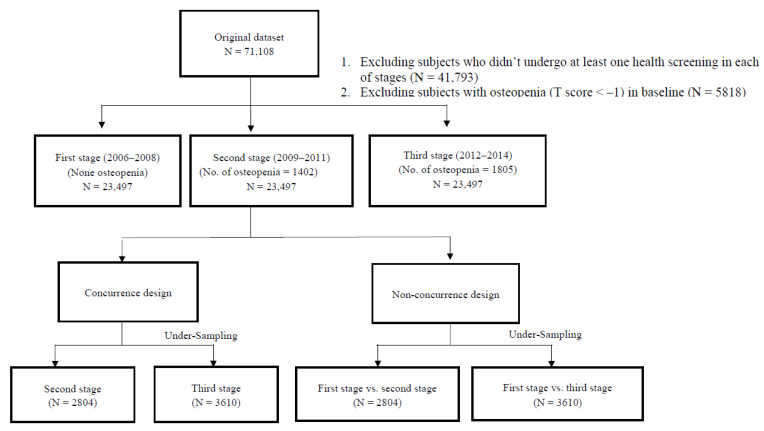
Flow chart of the data collection process used to identify study participants and define the analysis dataset.

**Figure 2 healthcare-09-00948-f002:**
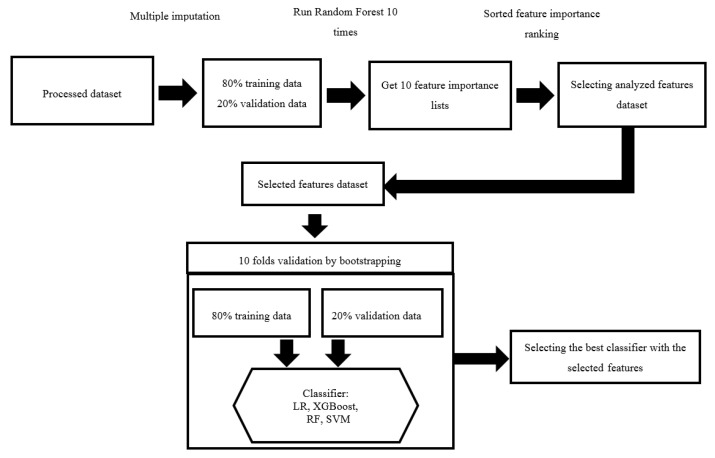
The process of feature selection and the application of machine learning algorithms.

**Figure 3 healthcare-09-00948-f003:**
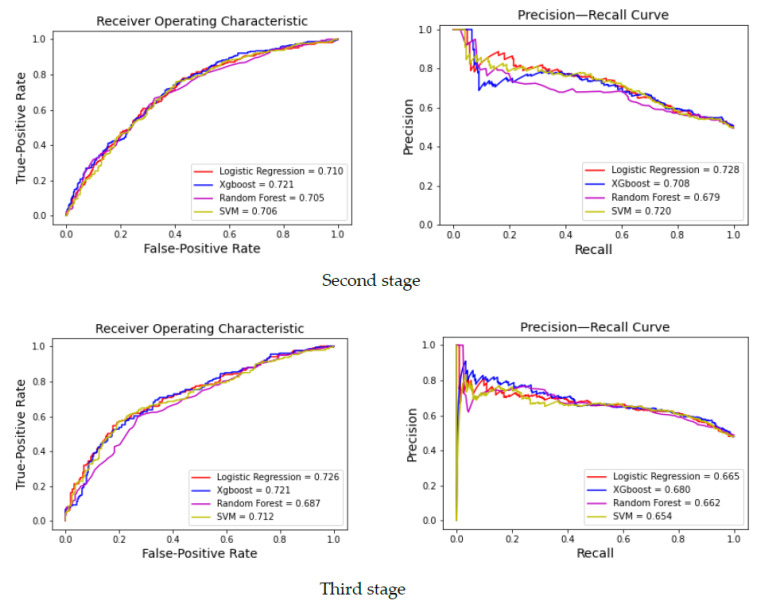
ROC and PRC curves for the machine learning models with concurrence designs.

**Figure 4 healthcare-09-00948-f004:**
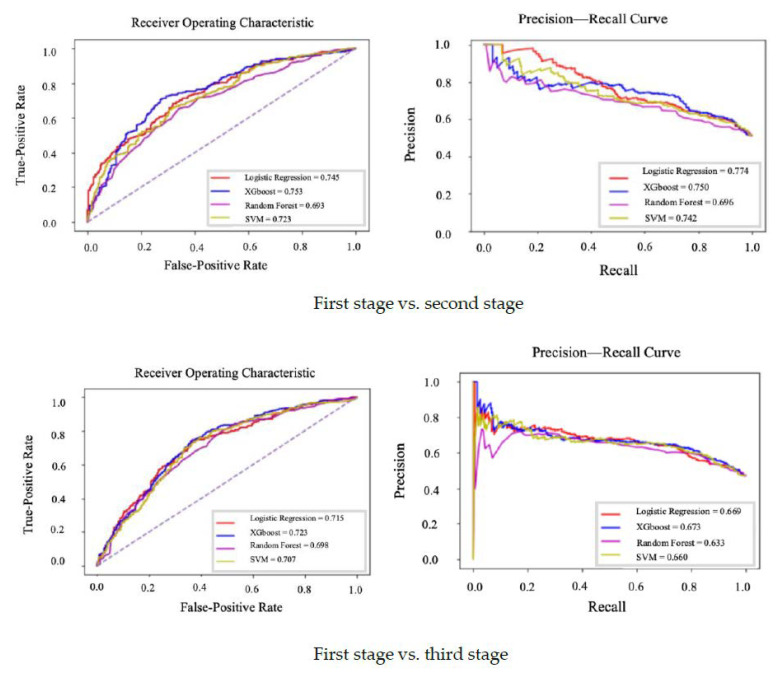
ROC and PRC curves for machine learning models with non-concurrence designs.

**Table 1 healthcare-09-00948-t001:** Robust feature importance ranking list.

Feature	Rank	Relative Importance
MetS score	1	1.000
Body mass index	2	0.959
Age	3	0.253
Education	4	0.243
Sweetened beverage	5	0.216
Milk intake	6	0.207
Income	7	0.194
Physical activity	8	0.187
Sleep	9	0.184
Occupation	10	0.162
Cheese intake	11	0.154
Sex	12	0.151
Smoke	13	0.133
Alcohol	14	0.127
Vitamin C/E intake	15	0.105
Calcium intake	16	0.103
Marital status	17	0.102

**Table 2 healthcare-09-00948-t002:** Explanatory variables related to osteopenia over the three study stages.

Characteristics	Participants in 2006–2008	Osteopenia in 2009–2011	Osteopenia in 2012–2014
	n (%)	n (%)	n (%)
Sex			
Male	13,012 (55.4)	953 (7.3)	1080 (8.3)
Female	10,485 (44.8)	449 (4.3)	725 (6.9)
Age (yrs)			
20–39	11,055 (47.0)	240 (2.9)	176 (3.0)
40–64	11,781 (50.1)	1029 (7.2)	1404 (8.7)
≥65	661 (2.8)	133 (14.0)	225 (16.4)
Marital status			
Unmarried	5163 (23.3)	211 (4.7)	258 (6.5)
Married	16,956 (76.7)	1100 (6.3)	1386 (7.8)
Education (yrs)			
<12	2178 (9.4)	289 (13.6)	346 (16.5)
12–15	10,529 (45.6)	635 (6.2)	777 (7.9)
≥16	10,397 (45.0)	444 (4.2)	586 (5.5)
Income (NTD/yr)			
<400,000	2676 (12.4)	226 (9.0)	297 (12.1)
400,000–799,999	5797 (26.8)	332 (6.4)	403 (8.4)
>800,000	13,174 (60.9)	699 (5.0)	899 (6.4)
Occupation			
Unemployed	3707 (17.5)	284 (7.5)	422 (10.8)
Managed	2562 (11.7)	150 (5.5)	183 (6.6)
Non-managed	15,557 (71.3)	815 (5.4)	970 (6.6)
Smoke (pack/day)			
None	18,545 (82.2)	1062 (5.6)	1503 (7.6)
≤1	3177 (14.1)	181 (6.6)	196 (7.6)
>1	839 (3.7)	71 (10.4)	57 (8.9)
Alcohol (cup/day)			
None	18,601 (83.9)	1041 (5.7)	1477 (7.6)
1	1726 (7.8)	94 (5.5)	140 (7.6)
≥2	1847 (8.3)	129 (7.2)	140 (7.8)
Chewing betel nut			
No	21,521 (93.8)	1208 (5.7)	1659 (7.6)
Yes	1428 (6.2)	93 (8.1)	102 (8.5)
Physical activity (hrs/wk)			
<1	9042 (39.6)	503 (5.4)	552 (6.7)
1–6	12,805 (56.1)	573 (6.1)	801 (7.8)
≥7	987 (4.3)	126 (7.3)	197 (10.8)
Sleep (hrs/day)			
<6	4523 (20.1)	369 (7.0)	524 (8.9)
6	16,467 (73.2)	676 (5.9)	845 (7.3)
≥7	1506 (6.7)	312 (5.1)	388 (6.9)
Vegetarian diet			
Yes	592 (2.5)	56 (8.2)	271 (7.9)
No	22,774 (97.5)	1330 (5.9)	1534 (7.6)
Sweetened beverage (cup/wk)			
None	7148 (30.8)	707 (7.2)	996 (8.8)
1–6	10,981 (47.3)	483 (5.0)	560 (6.4)
≥7	5067 (21.8)	176 (4.9)	189 (6.5)
Milk intake (cup/wk)			
None	11,545 (49.9)	701 (6.0)	871 (7.6)
1–6	9491 (41.0)	505 (5.6)	679 (7.2)
≥7	2093 (9.0)	158 (7.4)	187 (9.3)
Cheese intake (slice/wk)			
None	13,276 (57.5)	824 (6.3)	1119 (8.4)
1–6	9390 (40.7)	503 (5.3)	581 (6.4)
≥7	430 (1.9)	33 (6.3)	32 (6.9)
Vitamin C, E intake			
Yes	4180 (17.8)	175 (4.8)	271 (7.9)
No	19,312 (82.2)	1227 (6.2)	1534 (7.6)
Calcium intake			
Yes	3990 (17.0)	326 (8.6)	403 (12.0)
No	19,502 (93.0)	1076 (5.5)	1402 (7.0)
Hypertension medicine			
Yes	1399 (6.0)	138 (7.1)	226 (9.3)
No	22,093 (94.0)	1264 (5.9)	1579 (7.5)
Diabetes medicine			
Yes	440 (1.9)	47 (7.1)	72 (8.5)
No	23,052 (98.1)	1355 (5.9)	1733 (7.7)
Thyroid medicine			
Yes	252 (1.1)	21 (6.5)	27 (7.3)
No	23,240 (98.9)	1381 (6.0)	1778 (7.7)
Lipidemia medicine			
Yes	400 (1.7)	35 (5.7)	68 (8.1)
No	23,092 (98.3)	1367 (6.0)	1737 (7.7)
Hormone medicine			
Yes	272 (1.2)	18 (7.9)	15 (7.1)
No	23,220 (98.8)	1384 (5.9)	1790 (7.7)
Body mass index (sd)	23.25 (3.41)	22.79 (3.09)	22.85 (3.07)
MetS score (sd)	0.09 (1.02)	−0.22 (0.99)	−0.22 (0.94)

**Table 3 healthcare-09-00948-t003:** Model predictions of osteopenia in the second stage (2009–2011) using concurrent and non-concurrent features.

	Logistic Regression		XGBoost		Random Forest		SVM	
	Non-Concurrent	Concurrent	Non-Concurrent	Concurrent	Non-Concurrent	Concurrent	Non-Concurrent	Concurrent
Sensitivity	0.682	0.684	0.733	0.678	0.663	0.636	0.736	0.702
Specificity	0.648	0.681	0.689	0.672	0.623	0.636	0.575	0.632
Accuracy	0.665	0.683	0.711	0.675	0.643	0.636	0.658	0.667
Precision	0.655	0.694	0.713	0.694	0.650	0.631	0.646	0.651
ROC	0.726	0.745	0.753	0.721	0.693	0.687	0.723	0.712
PRC	0.728	0.774	0.750	0.708	0.696	0.697	0.742	0.720
F1	0.668	0.689	0.723	0.686	0.656	0.633	0.688	0.676

SVM: support vector machine; XGBoost: extreme gradient boosting; ROC: receiver operating characteristic curve; PRC: precision–recall curve; Non-concurrence indicates the prediction using the individual features from the first stage (2006–2008).

**Table 4 healthcare-09-00948-t004:** Model predictions of osteopenia in the third stage (2012–2014) using concurrent and non-concurrent features.

	Logistic Regression		XGBoost		Random Forest		SVM	
	Non-Concurrent	Concurrent	Non-Concurrent	Concurrent	Non-Concurrent	Concurrent	Non-Concurrent	Concurrent
Sensitivity	0.704	0.698	0.745	0.662	0.680	0.672	0.751	0.698
Specificity	0.646	0.620	0.633	0.657	0.622	0.660	0.600	0.627
Accuracy	0.673	0.657	0.686	0.659	0.650	0.666	0.669	0.661
Precision	0.640	0.628	0.645	0.639	0.617	0.645	0.624	0.632
ROC	0.715	0.710	0.723	0.721	0.698	0.705	0.707	0.706
PRC	0.669	0.665	0.673	0.680	0.633	0.662	0.660	0.654
F1	0.670	0.661	0.691	0.650	0.647	0.658	0.681	0.663

SVM: support vector machine; XGBoost: extreme gradient boosting; ROC: receiver operating characteristic curve; PRC: precision–recall curve; Non-concurrence indicates the prediction using the individual features from the first stage (2006–2008).

## Data Availability

Data presented in this study are not available on request from the corresponding author. Due to the General Data Protection Regulation, the data presented in this research are not publicly available.

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
