# Peer review of "Machine Learning to Predict the Progression of Bone Mass Loss Associated with Personal Characteristics and a Metabolic Syndrome Scoring Index"

_healthcare, 2021, doi:10.3390/healthcare9080948_

Round 1

Reviewer 1 Report

Line 224 - Can you elaborate on why this is a useful measure in this scenario

Line 234 - Good to clarify that it is an increase to 8.3 & 6.9% and not by those percentages

Line 297 - Could you add further information as to why the authors think the models are more effective in predicting short term outcomes?

Line 308 - Could also add that non-invasive screening is also key to success

Line 359 - Possible typographical error - Did the author mean genotypic variables?

Whilst the discussion section states which tools are more predictive than othes, it would be better to include in the main text HOW predictive each algorithms is and how much comparative to others used in the study. Furthermore, it would be good to discuss how the level of prediction given by these models would provide (x number of) patients with a better outcome than the current standard of care.

Author Response

Greeting

I want to thank you for your thoughtful comments regarding my manuscript.  I have replied to each comment point-by-point below in the cover letter and marked up using the “Track Changes” in the new uploaded manuscript.  I have attempted to address each of your comments.  The manuscript has been edited by a native English speaker as well. Please let me know if you have any questions and I look forward to your further review.

Reviewer 2 Report

Overall, this study try to use ML and data that includes SES in prediction. I have some major and some minor comments.

Please give some background on the differences between Concurrence and non-concurrence modeling and reference to existing publication in the field of healthcare.

Line 91-92: “Using a machine learning approach, this study developed a model to predict the loss of adult bone mass among a Taiwanese population using the novel MetS severity score and individual risk factors.”

  • What is meant by “novel MetS severity score” in this sentence. Is the study creating and validating a score system or applying ML to identify at-risk patients for loss of bone mass??

Line 99-101:  “A detailed description of the data collection and analysis of the resulting Major Longitudinal Health-check-up-based Population Database (MJLPD) is described in detail elsewhere [15,25].”

  • When checking ref. 15: “The data collection and analysis of the resulting Major Longitudinal Health-check-up-based Population Database (MJLPD) has been described in detail elsewhere in References [28,29]” ref. 28 is behind paywall; some information can be found in ref. 29. And ref. 29 is the same as ref. 25 in the main manuscript.
  • It is preferred to have all the relevant information within this paper, so the reader does not have go back to references which directs to other references ....

Line 117-119: “Due to a relatively low positive rate, the dataset was analyzed using a random undersampling (1:1 match) approach while applying machine learning models to mitigate the imbalance problem.”

  • Regarding the undersampling. This is acceptable approach, however, only when applied to the training dataset and not testing dataset. It is not clear how the study was designed. pleaser clarify.

Line 131: “All BMD reports were independently reviewed and coded by trained research physicians.”

  • If this person is a co-author(s), please indicate the initials in paranthesis

Line 138: “Using −1.0 SD as the cutoff point”

  • Is this cutoff arbitrary/data-driven or used by others before (please cite); indicate how this cutoff was determined.

Line 181: exclusion based on missingness ; line 182: imputation

  • Cause for selection bias (excluding patients because of missingness). Please indicate this as a study limitation.
  • Imputation procedure not clearly explained. Please provide more details and package used (if any).

Line 189: “the analysis ten times to catch data from the dataset.”

  • What is catch data?

Line 191: “forecasting the prediction included 17 of the 24”

  • Not clear how 17 variables were selected? What basis was the importance cut-off based upon? Please elaborate. Or ideally present the results using all the 24 features for comparison.

Line 198-200: “machine learning algorithms…”

  • Please spell out first time use of abbreviation (ex: RF)
  • In the modeling process, it is not clear if under sampling was done on the whole data or just on the training data. Testing dataset should not be affected by this artificial manipulation as it will generally produce better results (because of the prevalence) if is has 1:1 ratio.

Line 242: Table 1. Explanatory variables

  • Please provide p-value associated with the different variables and groups. This table is dense (very informative) however the reader will appreciate to have the p-value to quickly see where the significant difference are.

Line 246: “model via a testing dataset derived from the training dataset”

  • Not clear what is meant by “testing dataset derived from the training dataset”? please clarify/rephrase.

Table 3 and 4 and discussion

  • I am not convinced that the concurrent and non- concurrent strategy have significant differences in performance. please expand.

Line 295: MetS severity score

  • The authors talk about the “novel” MetS score, perhaps it would be important to have a short section in the manuscript to describe this score and its novelty more clearly.

Discussion / conclusion:

The study does reach to the level of "good" performance, but it is really borderline. It would have been best if authors did more optimization, in terms of up-sampling, feature selection using various methods, or better imputation, or even using other ML algorithms. There is definitely room to improve. The following study (https://www.mdpi.com/2077-0383/10/6/1286/htm ) includes a number of steps in model optimization but that is just one example, there are plenty of very nicely done ML work in the literature.

Various Grammatical and typos in the text. Please consider a careful read before resubmission.

Author Response

(The authors gave the same response as above.)

Round 2

Reviewer 2 Report

Thank you for the edits. However, not all the points were sufficiently addressed. Furthermore, I strongly believe more model optimization steps needs to be done before I can recommend accepting this manuscript; with this level of performance and the ability to run a lot of these optimization easily on the data, I recommend authors to not leave that for future studies but try to improve the model performance as they also acknowledged (see below). 

"We agree with your suggestion. Other than the modelling strategy the study used, there are several different algorithms (e.g. gradient boosting machine, decision tree etc.), samplings, and feature selection such as data-driven approaches [Wiener, M. Classification and Regression by randomForest. R News 2002, 2, 18–22.; Friedman, J.H.; Hastie, T.; Tibshirani, R. Regularization Paths for Generalized Linear Models via Coordinate Descent. J. Stat. Softw. 2010, 33, 1–22. ; Chawla, N.V.; Bowyer, K.W.; Hall, L.O.; Kegelmeyer, W.P. SMOTE: Synthetic minority over-sampling technique. J. Artif. Intell. Res. 2002, 16, 321–357.] can be used to optimize the performance of machine learning. Further studies with those addition approaches may be needed. We add the statements in the third paragraph of the Discussion section."

Author Response

Thank you for your comments. As we performed additional analyses under varying approaches, the results showed that synthetic minority oversampling technique (SMOTE) could be used to optimize the performance of machine learning (Table S1 to S3). Future studies with the approach may be applicable. However, caution should be exercised to prevent adding increased uncertainty, especially in regards to a sample with a small number of examples of minority class or a non-continuous feature space. The sample size of the dataset generated by the SMOTE is as over ten times as the dataset we analyzed initially. We add the statement in the Discussion section. The manuscript has been edited by a native English speaker as well.
